# Implicit Generation and Modeling with Energy-Based Models

**Yilun Du** *
MIT CSAIL

**Igor Mordatch**
Google Brain

## Abstract

Energy based models (EBMs) are appealing due to their generality and simplicity in likelihood modeling, but have been traditionally difficult to train. We present techniques to scale MCMC based EBM training on continuous neural networks, and we show its success on the high-dimensional data domains of ImageNet32x32, ImageNet128x128, CIFAR-10, and robotic hand trajectories, achieving better samples than other likelihood models and nearing the performance of contemporary GAN approaches, while covering all modes of the data. We highlight some unique capabilities of implicit generation such as compositionality and corrupt image reconstruction and inpainting. Finally, we show that EBMs are useful models across a wide variety of tasks, achieving state-of-the-art out-of-distribution classification, adversarially robust classification, state-of-the-art continual online class learning, and coherent long term predicted trajectory rollouts.

## 1 Introduction

Learning models of the data distribution and generating samples are important problems in machine learning for which a number of methods have been proposed, such as Variational Autoencoders (VAEs) [Kingma and Welling, 2014] and Generative Adversarial Networks (GANs) [Goodfellow et al., 2014].In this work, we advocate for continuous energy-based models (EBMs), represented as neural networks, for generative modeling tasks and as a building block for a wide variety of tasks. These models aim to learn an energy function $E(\mathbf{x})$ that assigns low energy values to inputs $\mathbf{x}$ in the data distribution and high energy values to other inputs. Importantly, they allow the use of an *implicit* sample generation procedure, where sample $\mathbf{x}$ is found from $\mathbf{x} \sim e^{-E(\mathbf{x})}$ through MCMC sampling. Combining implicit sampling with energy-based models for generative modeling has a number of conceptual advantages compared to methods such as VAEs and GANs which use explicit functions to generate samples:

**Simplicity and Stability:** An EBM is the only object that needs to be trained and designed. Separate networks are not tuned to ensure balance (for example, [He et al., 2019] point out unbalanced training can result in posterior collapse in VAEs or poor performance in GANs [Kurach et al., 2018]).

**Sharing of Statistical Strength:** Since the EBM is the only trained object, it requires fewer model parameters than approaches that use multiple networks. More importantly, the model being concentrated in a single network allows the training process to develop a shared set of features as opposed to developing them redundantly in separate networks.

**Adaptive Computation Time:** Implicit sample generation in our work is an iterative stochastic optimization process, which allows for a trade-off between generation quality and computation time.

This allows for a system that can make fast coarse guesses or more deliberate inferences by running the optimization process longer. It also allows for refinement of external guesses.

**Flexibility Of Generation:** The power of an explicit generator network can become a bottleneck on the generation quality. For example, VAEs and flow-based models are bound by the manifold structure of the prior distribution and consequently have issues modeling discontinuous data manifolds, often assigning probability mass to areas unwarranted by the data. EBMs avoid this issue by directly modeling particular regions as high or lower energy.

**Compositionality:** If we think of energy functions as costs for a certain goals or constraints, summation of two or more energies corresponds to satisfying all their goals or constraints [Mnih and Hinton, 2004, Haarnoja et al., 2017]. While such composition is simple for energy functions (or product of experts [Hinton, 1999]), it induces complex changes to the generator that may be difficult to capture with explicit generator networks.

Despite these advantages, energy-based models with implicit generation have been difficult to use on complex high-dimensional data domains. In this work, we use Langevin dynamics [Welling and Teh, 2011], which uses gradient information for effective sampling and initializes chains from random noise for more mixing. We further maintain a replay buffer of past samples (similarly to [Tieleman, 2008] or [Mnih et al., 2013]) and use them to initialize Langevin dynamics to allow mixing between chains. An overview of our approach is presented in Figure 1.

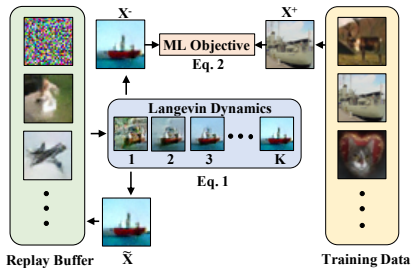

Figure 1: Overview of our method and the interrelationship of the components involved.

Empirically, we show that energy-based models trained on CIFAR-10 or ImageNet image datasets generate higher quality image samples than likelihood models and nearing that of contemporary GANs approaches, while not suffering from mode collapse. The models exhibit properties such as correctly assigning lower likelihood to out-of-distribution images than other methods (no spurious modes) and generating diverse plausible image completions (covering all data modes). Implicit generation allows our models to naturally denoise or inpaint corrupted images, convert general images to an image from a specific class, and generate samples that are compositions of multiple independent models.

Our contributions in this work are threefold. Firstly, we present an algorithm and techniques for training energy-based models that scale to challenging high-dimensional domains. Secondly, we highlight unique properties of energy-based models with implicit generation, such as compositionality and automatic decorruption and inpainting. Finally, we show that energy-based models are useful across a series of domains, on tasks such as out-of-distribution generalization, adversarially robust classification, multi-step trajectory prediction and online learning.

## 2 Related Work

Energy-based models (EBMs) have a long history in machine learning. Ackley et al. [1985], Hinton [2006], Salakhutdinov and Hinton [2009] proposed latent based EBMs where energy is represented as a composition of latent and observable variables. In contrast Mnih and Hinton [2004], Hinton et al. [2006] proposed EBMs where inputs are directly mapped to outputs, a structure we follow. We refer readers to [LeCun et al., 2006] for a comprehensive tutorial on energy models.

The primary difficulty in training EBMs comes from effectively estimating and sampling the partition function. One approach to train energy based models is sample the partition function through amortized generation. Kim and Bengio [2016], Zhao et al. [2016], Haarnoja et al. [2017], Kumar et al. [2019] propose learning a separate network to generate samples, which makes these methods closely connected to GANs [Finn et al., 2016], but these methods do not have the advantages of implicit sampling noted in the introduction. Furthermore, amortized generation is prone to mode collapse, especially when training the sampling network without an entropy term which is often approximated or ignored.

An alternative approach is to use MCMC sampling to estimate the partition function. This has an advantage of provable mode exploration and allows the benefits of implicit generation listed in the

introduction. Hinton [2006] proposed Contrastive Divergence, which uses gradient free MCMC chains initialized from training data to estimate the partition function. Similarly, Salakhutdinov and Hinton [2009] apply contrastive divergence, while Tieleman [2008] proposes PCD, which propagates MCMC chains throughout training. By contrast, we initialize chains from random noise, allowing each mode of the model to be visited with equal probability. But initialization from random noise comes at a cost of longer mixing times. As a result we use Gradient based MCMC (Langevin Dynamics) for more efficient sampling and to offset the increase of mixing time which was also studied previously in [Teh et al., 2003, Xie et al., 2016]. We note that HMC [Neal, 2011] may be an even more efficient gradient algorithm for MCMC sampling, though we found Langevin Dynamics to be more stable. To allow gradient based MCMC, we use continuous inputs, while most approaches have used discrete inputs. We build on idea of PCD and maintain a replay buffer of past samples to additionally reduce mixing times.

# 3 Energy-Based Models and Sampling

Given a datapoint $\mathbf{x}$, let $E_\theta(\mathbf{x}) \in \mathbb{R}$ be the energy function. In our work this function is represented by a deep neural network parameterized by weights $\theta$. The energy function defines a probability distribution via the Boltzmann distribution $p_\theta(\mathbf{x}) = \frac{\exp(-E_\theta(\mathbf{x}))}{Z(\theta)}$, where $Z(\theta) = \int \exp(-E_\theta(\mathbf{x}))d\mathbf{x}$ denotes the partition function. Generating samples from this distribution is challenging, with previous work relying on MCMC methods such as random walk or Gibbs sampling [Hinton, 2006]. These methods have long mixing times, especially for high-dimensional complex data such as images. To improve the mixing time of the sampling procedure, we use Langevin dynamics which makes use of the gradient of the energy function to undergo sampling

$$\tilde{\mathbf{x}}^k = \tilde{\mathbf{x}}^{k-1} - \frac{\lambda}{2}\nabla_{\mathbf{x}}E_\theta(\tilde{\mathbf{x}}^{\smile 1}) + \omega^k, \ \omega^k \sim \mathcal{N}(0,\lambda) \quad (1)$$

where we let the above iterative procedure define a distribution $q_\theta$ such that $\tilde{\mathbf{x}}^K \sim q_\theta$. As shown by Welling and Teh [2011] as $K \to \infty$ and $\lambda \to 0$ then $q_\theta \to p_\theta$ and this procedure generates samples from the distribution defined by the energy function. Thus, samples are generated implicitly[†] by the energy function $E$ as opposed to being explicitly generated by a feedforward network.

In the domain of images, if the energy network has a convolutional architecture, energy gradient $\nabla_{\mathbf{x}}E$ in (1) conveniently has a deconvolutional architecture. Thus it mirrors a typical image generator network architecture, but without it needing to be explicitly designed or balanced. We take two views of the energy function $E$: firstly, it is an object that defines a probability distribution over data and secondly it defines an implicit generator via (1).

## 3.1 Maximum Likelihood Training

We want the distribution defined by $E$ to model the data distribution $p_D$, which we do by minimizing the negative log likelihood of the data $\mathcal{L}_{\mathrm{ML}}(\theta) = \mathbb{E}_{\mathbf{x} \sim p_D}\left[-\log p_\theta(\mathbf{x})\right]$ where $-\log p_\theta(\mathbf{x}) = E_\theta(\mathbf{x}) - \log Z(\theta)$. This objective is known to have the gradient (see [Turner, 2005] for derivation) $\nabla_\theta \mathcal{L}_{\mathrm{ML}} = \mathbb{E}_{\mathbf{x}^+ \sim p_D}\left[\nabla_\theta E_\theta(\mathbf{x}^+)\right] - \mathbb{E}_{\mathbf{x}^- \sim p_\theta}\left[\nabla_\theta E_\theta(\mathbf{x}^-)\right]$. Intuitively, this gradient decreases energy of the positive data samples $\mathbf{x}^+$, while increasing the energy of the negative samples $\mathbf{x}^-$ from the model $p_\theta$. We rely on Langevin dynamics in (1) to generate $q_\theta$ as an approximation of $p_\theta$:

$$\nabla_\theta \mathcal{L}_{\mathrm{ML}} \approx \mathbb{E}_{\mathbf{x}^+ \sim p_D}\left[\nabla_\theta E_\theta(\mathbf{x}^+)\right] - \mathbb{E}_{\mathbf{x}^- \sim q_\theta}\left[\nabla_\theta E_\theta(\mathbf{x}^-)\right]. \quad (2)$$

This is similar to the gradient of the Wasserstein GAN objective [Arjovsky et al., 2017], but with an implicit MCMC generating procedure and no gradient through sampling. This lack of gradient is important as it controls between the diversity in likelihood models and the mode collapse in GANs.

The approximation in (2) is exact when Langevin dynamics generates samples from $p$, which happens after a sufficient number of steps (mixing time). We show in the supplement that $p_d$ and $q$ appear to match each other in distribution, showing evidence that $p$ matches $q$. We note that even in cases when a particular chain does not fully mix, since our initial proposal distribution is a uniform distribution, all modes are still equally likely to be explored.

---

[†]Deterministic case of procedure in (1) is $\mathbf{x} = \arg\min E(\mathbf{x})$, which makes connection to implicit functions more clear.

## 3.2 Sample Replay Buffer

Langevin dynamics does not place restrictions on sample initialization $\tilde{\mathbf{x}}^0$ given sufficient sampling steps. However initialization plays an crucial role in mixing time. Persistent Contrastive Divergence (PCD) [Tieleman, 2008] maintains a single persistent chain to improve mixing and sample quality. We use a sample replay buffer $\mathcal{B}$ in which we store past generated samples $\tilde{\mathbf{x}}$ and use either these samples or uniform noise to initialize Langevin dynamics procedure. This has the benefit of continuing to refine past samples, further increasing number of sampling steps $K$ as well as sample diversity. In all our experiments, we sample from $\mathcal{B}$ 95% of the time and from uniform noise otherwise.

## 3.3 Regularization and Algorithm

Arbitrary energy models can have sharp changes in gradients that can make sampling with Langevin dynamics unstable. We found that constraining the Lipschitz constant of the energy network can ameliorate these issues. To constrain the Lipschitz constant, we follow the method of [Miyato et al., 2018] and add spectral normalization to all layers of the model. Additionally, we found it useful to weakly L2 regularize energy magnitudes for both positive and negative samples during training, as otherwise while the difference between positive and negative samples was preserved, the actual values would fluctuate to numerically unstable values. Both forms of regularization also serve to ensure that partition function is integrable over the domain of the input, with spectral normalization ensuring smoothness and L2 coefficient bounding the magnitude of the unnormalized distribution. We present the algorithm below, where $\Omega(\cdot)$ indicates the stop gradient operator.

---

**Algorithm 1** Energy training algorithm

**Input:** data dist. $p_D(\mathbf{x})$, step size $\lambda$, number of steps $K$
$\mathcal{B} \leftarrow \varnothing$
**while** not converged **do**
    $\mathbf{x}_i^+ \sim p_D$
    $\mathbf{x}_i^0 \sim \mathcal{B}$ with 95% probability and $\mathcal{U}$ otherwise

    ▷ *Generate sample from $q_\theta$ via Langevin dynamics:*
    **for** sample step $k = 1$ to $K$ **do**
        $\tilde{\mathbf{x}}^k \leftarrow \tilde{\mathbf{x}}^{k-1} - \nabla_{\mathbf{x}} E_\theta(\tilde{\mathbf{x}}^{k-1}) + \omega, \quad \omega \sim \mathcal{N}(0, \sigma)$
    **end for**
    $\mathbf{x}_i^- = \Omega(\tilde{\mathbf{x}}_i^k)$

    ▷ *Optimize objective $\alpha \mathcal{L}_2 + \mathcal{L}_{ML}$ wrt $\theta$:*
    $\Delta\theta \leftarrow \nabla_\theta \frac{1}{N} \sum_i \alpha(E_\theta(\mathbf{x}_i^+)^2 + E_\theta(\mathbf{x}_i^-)^2) + E_\theta(\mathbf{x}_i^+) - E_\theta(\mathbf{x}_i^-)$
    Update $\theta$ based on $\Delta\theta$ using Adam optimizer

    $\mathcal{B} \leftarrow \mathcal{B} \cup \tilde{\mathbf{x}}_i$
**end while**

---

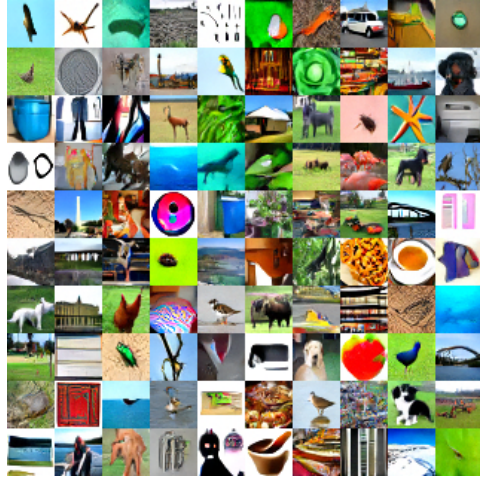

Figure 2: Conditional ImageNet32x32 EBM samples

## 4 Image Modeling

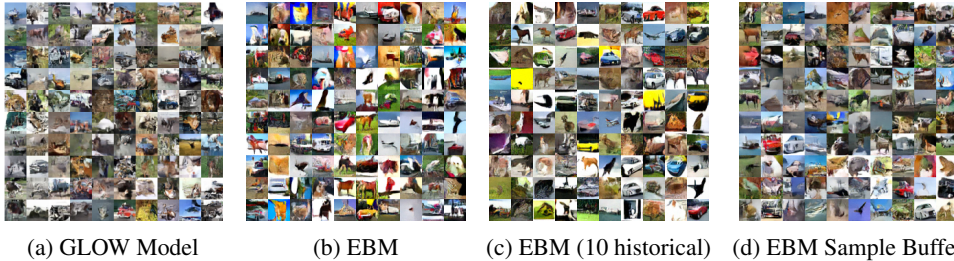

(a) GLOW Model      (b) EBM      (c) EBM (10 historical)    (d) EBM Sample Buffer

Figure 3: Comparison of image generation techniques on unconditional CIFAR-10 dataset.

In this section, we show that EBMs are effective generative models for images. We show EBMs are able to generate high fidelity images and exhibit mode coverage on CIFAR-10 and ImageNet.

We further show EBMs exhibit adversarial robustness and better out-of-distribution behavior than other likelihood models. Our model is based on the ResNet architecture (using conditional gains and biases per class [Dumoulin et al.] for conditional models) with details in the supplement. We present sensitivity analysis, likelihoods, and ablations in the supplement in A.4. We provide a comparison between EBMs and other likelihood models in A.5. Overall, we find that EBMs are both more parameter/computationally efficient than likelihood models, though worse than GANs.

## 4.1 Image Generation

We show unconditional CIFAR-10 images in Figure 3, with comparisons to GLOW [Kingma and Dhariwal, 2018], and conditional ImageNet32x32 images in Figure 2. We provide qualitative images of ImageNet128x128 and other visualizations in A.1.

| Model | Inception* | FID |
|---|---|---|
| **CIFAR-10 Unconditional** | | |
| PixelCNN [Van Oord et al., 2016] | 4.60 | 65.93 |
| PixelIQN [Ostrovski et al., 2018] | 5.29 | 49.46 |
| EBM (single) | 6.02 | 40.58 |
| DCGAN [Radford et al., 2016] | 6.40 | 37.11 |
| WGAN + GP [Gulrajani et al., 2017] | 6.50 | 36.4 |
| EBM (10 historical ensemble) | 6.78 | 38.2 |
| SNGAN [Miyato et al., 2018] | **8.22** | 21.7 |
| **CIFAR-10 Conditional** | | |
| Improved GAN | 8.09 | - |
| EBM (single) | 8.30 | 37.9 |
| Spectral Normalization GAN | **8.59** | 25.5 |
| **ImageNet 32x32 Conditional** | | |
| PixelCNN | 8.33 | 33.27 |
| PixelIQN | 10.18 | 22.99 |
| EBM (single) | **18.22** | **14.31** |
| **ImageNet 128x128 Conditional** | | |
| ACGAN [Odena et al., 2017] | 28.5 | - |
| EBM* (single) | 28.6 | 43.7 |
| SNGAN | **36.8** | 27.62 |

Figure 4: Table of Inception and FID scores for ImageNet32x32 and CIFAR-10. Quantitative numbers for ImageNet32x32 from [Ostrovski et al., 2018]. (*) We use Inception Score (from original OpenAI repo) to compare with legacy models, but strongly encourage future work to compare soley with FID score, since Langevin Dynamics converges to minima that artificially inflate Inception Score. (**) conditional EBM models for 128x128 are smaller than those in SNGAN.

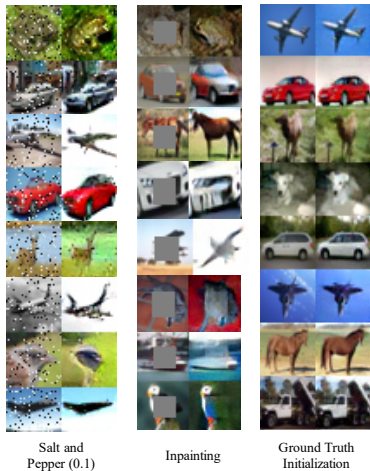

Salt and Pepper (0.1)    Inpainting    Ground Truth Initialization

Figure 5: EBM image restoration on images in the **test** set via MCMC. The right column shows failure (approx. 10% objects change with ground truth initialization and 30% of objects change in salt/pepper corruption or inpainting. Bottom two rows shows worst case of change.)

We quantitatively evaluate image quality of EBMs with Inception score [Salimans et al., 2016] and FID score [Heusel et al., 2017] in Table 4. Overall we obtain significantly better scores than likelihood models PixelCNN and PixelIQN, but worse than SNGAN [Miyato et al., 2018]. We found that in the unconditional case, mode exploration with Langevin took a very long time, so we also experimented in *EBM (10 historical ensemble)* with sampling joint from the last 10 snapshots of the model. At training time, extensive exploration is ensured with the replay buffer (Figure 3d). Our models have similar number of parameters to SNGAN, but we believe that significantly more parameters may be necessary to generate high fidelity images with mode coverage. On ImageNet128x128, due to computational constraints, we train a smaller network than SNGAN and do not train to convergence.

## 4.2 Mode Evaluation

We evaluate over-fitting and mode coverage in EBMs. To test over-fitting, we plotted histogram of energies for CIFAR-10 train and test dataset in Figure 11 and note almost identical curves. In the supplement, we show that the nearest neighbor of generated images are not identical to images in

the training dataset. To test mode coverage in EBMs, we investigate MCMC sampling on corrupted CIFAR-10 test images. Since Langevin dynamics is known to mix slowly [Neal, 2011] and reach local minima, we believe that good denoising after limited number of steps of sampling indicates probability modes at respective test images. Similarly, lack of movement from a ground truth test image initialization after the same number of steps likely indicates probability mode at the test image. In Figure 5, we find that if we initialize sampling with images from the test set, images do not move significantly. However, under the same number of steps, Figure 5 shows that we are able to reliably decorrupt masked and salt and pepper corrupted images, indicating good mode coverage. We note that large number of steps of sampling lead to more saturated images, which are due to sampling low temperature modes, which are saturated across likelihood models (see appendix). In comparison, GANs have been shown to miss many modes of data and cannot reliably reconstruct many different test images [Yeh et al.]. We note that such decorruption behavior is a nice property of implicit generation without need of explicit knowledge of corrupted pixels.

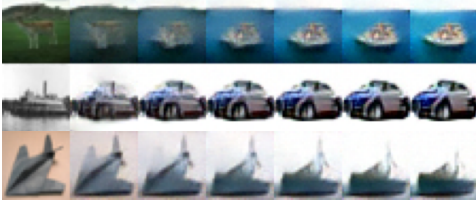

Figure 6: Illustration of cross-class implicit sampling on a conditional EBM. The EBM is conditioned on a particular class but is initialized with an image from a separate class.

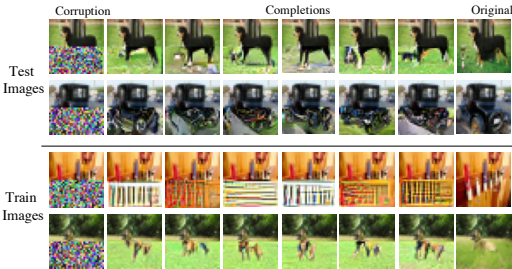

Figure 7: Illustration of image completions on conditional ImageNet model. Our models exhibit diversity in inpainting.

Another common test for mode coverage and overfitting is masked inpainting [Van Oord et al., 2016]. In Figure 7, we mask out the bottom half of ImageNet images and test the ability to sample the masked pixels, while fixing the value of unmasked pixels. Running Langevin dynamics on the images, we find diversity of completions on train/test images, indicating low overfitting on training set and diversity characterized by likelihood models. Furthermore initializing sampling of a class conditional EBM with images from images from another class, we can further test for presence of probability modes at images far away from the those seen in training. We find in Figure 6 that sampling on such images using an EBM is able to generate images of the target class, indicating semantically meaningful modes of probability even far away from the training distribution.

## 4.3 Adversarial Robustness

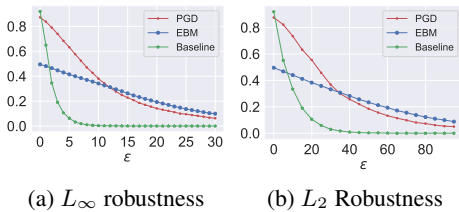

(a) $L_\infty$ robustness    (b) $L_2$ Robustness

Figure 8: $\epsilon$ plots under $L_\infty$ and $L_2$ attacks of conditional EBMs as compared to PGD trained models in [Madry et al., 2017] and a baseline Wide ResNet18.

We show conditional EBMs exhibit adversarial robustness on CIFAR-10 classification, **without explicit** adversarial training. To compute logits for classification, we compute the negative energy of the image in each class. Our model, without fine-tuning, achieves an accuracy of 49.6%. Figure 8 shows adversarial robustness curves. We ran 20 steps of PGD as in [Madry et al., 2017], on the above logits. To undergo classification, we then ran 10 steps sampling initialized from the starting image (with a bounded deviation of 0.03) from each conditional model, and then classified using the lowest energy conditional class. We found that running PGD incorporating sampling was less successful than without. Overall we find in Figure 8 that EBMs are very robust to adversarial perturbations and outperforms the SOTA $L_\infty$ model in [Madry et al., 2017] on $L_\infty$ attacks with $\epsilon > 13$.

## 4.4 Out-of-Distribution Generalization

We show EBMs exhibit better out-of-distribution (OOD) detection than other likelihood models. Such a task requires models to have high likelihood on the data manifold and low likelihood at all

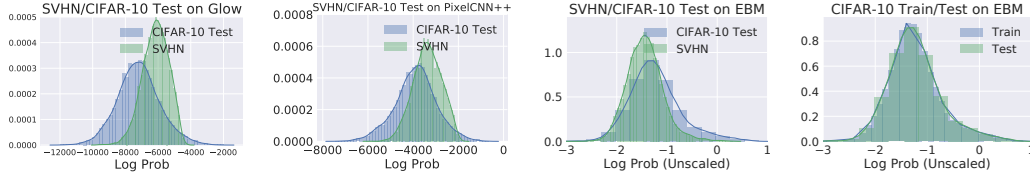

Figure 11: Histogram of relative likelihoods for various datasets for Glow, PixelCNN++ and EBM models

other locations and can be viewed as a proxy of log likelihood. Surprisingly, Nalisnick et al. [2019] found likelihood models such as VAE, PixelCNN, and Glow models, are unable to distinguish data assign higher likelihood to many OOD images. We constructed our OOD metric following following [Hendrycks and Gimpel, 2016] using Area Under the ROC Curve (AUROC) scores computed based on classifying CIFAR-10 test images from other OOD images using relative log likelihoods. We use SVHN, Textures [Cimpoi et al., 2014], monochrome images, uniform noise and interpolations of separate CIFAR-10 images as OOD distributions. We provide examples of OOD images in Figure 9. We found that our proposed OOD metric correlated well with training progress in EBMs.

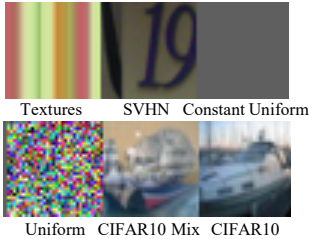

Figure 9: Illustration of images from each of the out of distribution dataset.

| Model | PixelCNN++ | Glow | EBM (ours) |
|---|---|---|---|
| SVHN | 0.32 | 0.24 | **0.63** |
| Textures | 0.33 | 0.27 | **0.48** |
| Constant Uniform | 0.0 | 0.0 | **0.30** |
| Uniform | 1.0 | 1.0 | **1.0** |
| CIFAR10 Interpolation | **0.71** | 0.59 | 0.70 |
| Average | 0.47 | 0.42 | **0.62** |

Figure 10: AUROC scores of out of distribution classification on different datasets. Only our model gets better than chance classification.

In Table 10, unconditional EBMs perform **significantly better** out-of-distribution than other auto-regressive and flow generative models and have OOD scores of 0.62 while the closest, PixelCNN++, has a OOD score of 0.47. We provide histograms of relative likelihoods for SVHN in Figure 11 which are also discussed in [Nalisnick et al., 2019, Hendrycks et al., 2018]. We believe that the reason for better generalization is two-fold. First, we believe that the negative sampling procedure in EBMs helps eliminate spurious minima. Second, we believe EBMs have a flexible structure that allows global context when estimating probability without imposing constraints on latent variable structure. In contrast, auto-regressive models model likelihood sequentially, which makes global coherence difficult. In a different vein, flow based models must apply continuous transformations onto a continuous connected probability distribution which makes it very difficult to model disconnected modes, and thus assign spurious density to connections between modes.

## 5 Trajectory Modeling

We show that EBMs generate and generalize well in the different domain of trajectory modeling. We train EBMs to model dynamics of a simulated robot hand manipulating a free cube object [OpenAI, 2018]. We generated 200,000 different trajectories of length 100, from a trained policy (with every 4th action set to a random action for diversity), with a 90-10 train-test split. Models are trained to predict positions of all joints in the hand and orientation and position of the cube one step in the future. We test performance by evaluating many step roll-outs of self-predicted trajectories.

### 5.1 Training Setup and Metrics

We compare EBM models to feedforward models (FC), both of which are composed of 3 layers of 128 hidden units. We apply spectral normalization to FC to prevent multi-step explosion. We evaluate multi-step trajectories by computing Frechet Distance [Dowson and Landau, 1982] between predicted and ground distributions across all states at timestep $t$. We found this metric was a better metric of trajectories than multi-step MSE due to accumulation of error.

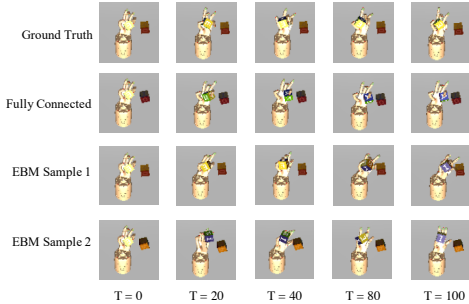

Figure 12: Views of hand manipulation trajectories generated unconditionally from the same state(1st frame).

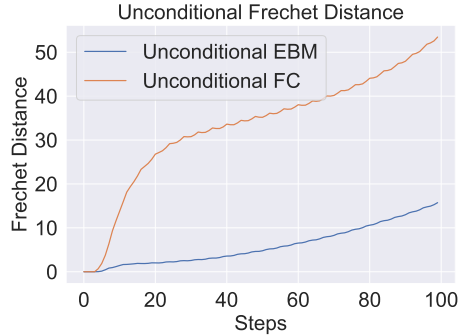

Figure 13: Conditional and Unconditional Modeling of Hand Manipulation through Frechet Distance

## 5.2 Multi-Step Trajectory Generation

We evaluated EBMs for both action conditional and unconditional prediction of multi-step rollouts. Quantitatively, by computing the average Frechet distance across all time-steps, unconditional EBM have value 5.96 while unconditional FC networks have a value of 33.28. Conditional EBM have value 8.97 while a conditional FC has value 19.75. We provide plots of Frechet distance over time in Figure 13. In Figure 13, we observe that for unconditional hand modeling in a FC network, the Frechet distance increases dramatically in the first several time steps. Qualitatively, we found that the same FC networks stop predicting hand movement after several several steps as demonstrated in Figure 12. In contrast, Frechet distance increases slowly for unconditional EBMs. The unconditional models are able to represent multimodal transitions such as different types of cube rotation and Figure 12 shows that the unconditional EBMs generate diverse realistic trajectories.

## 6 Online Learning

| Method | Accuracy |
|---|---|
| EWC [Kirkpatrick et al., 2017] | 19.80 (0.05) |
| SI [Zenke et al., 2017] | 19.67 (0.09) |
| NAS [Schwarz et al., 2018] | 19.52 (0.29) |
| LwF [Li and Snavely, 2018] | 24.17 (0.33) |
| VAE | 40.04 (1.31) |
| EBM (ours) | **64.99** (4.27) |

Table 1: Comparison of various continual learning benchmarks. Values averaged acrossed 10 seeds reported as mean (standard deviation).

We find that EBMs also perform well in continual learning. We evaluate incremental class learning on the Split MNIST task proposed in [Farquhar and Gal, 2018]. The task evaluates overall MNIST digit classification accuracy given 5 sequential training tasks of disjoint pairs of digits. We train a conditional EBM with 2 layers of 400 hidden units work and compare with a generative conditional VAE baseline with both encoder/decoder having 2 layers of 400 hidden units. Additional training details are covered in the appendix. We train the generative models to represent the joint distribution of images and labels and classify based off the lowest energy label. Hsu et al. [2018] analyzed common continual learning algorithms such as EWC [Kirkpatrick et al., 2017], SI [Zenke et al., 2017] and NAS [Schwarz et al., 2018] and find they obtain performance around 20%. LwF [Li and Snavely, 2018] performed the best with performance of $24.17 \pm 0.33$, where all architectures use 2 layers of 400 hidden units. However, since each new task introduces two new MNIST digits, a test accuracy of around 20% indicates complete forgetting of previous tasks. In contrast, we found continual EBM training obtains **significantly higher** performance of $64.99 \pm 4.27$. All experiments were run with 10 seeds.

A crucial difference is that negative training in EBMs only locally "forgets" information corresponding to negative samples. Thus, when new classes are seen, negative samples are conditioned on the new class, and the EBM only forgets unlikely data from the new class. In contrast, the cross entropy objective used to train common continual learning algorithms down-weights the likelihood of all classes not seen. We can apply this insight on other generative models, by maximizing the likelihood of a class conditional model at train time and then using the highest likelihood class as classification results. We ran such a baseline using a VAE and obtained a performance of $40.04 \pm 1.31$, which is higher than other continual learning algorithms but less than that in a EBM.

# 7 Compositional Generation



Figure 14: A 2D example of combining EBMs through summation and the resulting sampling trajectories.

Finally, we show compositionality through implicit generation in EBMs. Consider a set of conditional EBMs for separate independent latents. Sampling through the joint distribution on all latents is represented by generation on an EBM that is the sum of each conditional EBM [Hinton, 1999] and corresponds to a product of experts model. As seen in Figure 14, summation naturally allows composition of EBMs. We sample from joint conditional distribution through Langevin dynamics sequentially from each model.

We conduct our experiments on the dSprites dataset [Higgins et al., 2017], which consists of all possible images of an object (square, circle or heart) varied by scale, position, rotation with labeled latents. We trained conditional EBMs for each latent and found that scale, position and rotation worked well. The latent for shape was learned poorly, and we found that even our unconditional models were not able to reliably generate different shapes which was also found in [Higgins et al., 2017]. We show some results on CelebA in A.6.

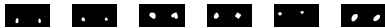

Figure 15: Samples from joint distribution of 4 independent conditional EBMs on scale, position, rotation and shape (left panel) with associated ground truth rendering (right panel).

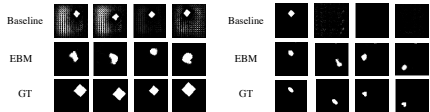

Figure 16: GT = Ground Truth. Images of cross product generalization of size-position (left panel) and shape-position (right panel).

**Joint Conditioning**   In Figure 15, we provide generated images from joint conditional sampling. Under such sampling we are able to generate images very close to ground truth for all classes with exception of shape. This result also demonstrates mode coverage across all data.

**Zero-Shot Cross Product Generalization**   We evaluate the ability of EBMs to generalize to novel combinations of latents. We generate three datasets, D1: different size squares at a central position, D2: smallest size square at each location, D3: different shapes at the center position. We evaluate size-position generalization by training independent energy functions on D1 and D2, and test on generating different size squares at all positions. We similarly evaluate shape-position generalization for D2 and D3. We generate samples at novel combinations by sampling from the summation of energy functions (we first finetune the summation energy to generate both training datasets using a KL term defined in the appendix). We compare against a joint conditional model baseline.

We present results of generalization in Figure 16. In the left panel of Figure 16, we find the EBMs are able to generalize to different sizes at different position (albeit with loss in sample quality) while a conditional model ignores the size latent and generates only images seen in the training. In the right panel of Figure 16, we found that EBMs are able to generalize to combinations of shape and position by creating a distinctive shape for each conditioned shape latent at different positions (though the generated shape doesn't match the shape of the original shape latent), while a baseline is unable to generate samples. We believe the compositional nature of EBMs is crucial to generalize in this task.

# 8 Conclusion

We have presented a series of techniques to scale up EBM training. We further show unique benefits of implicit generation and EBMs and believe there are many further directions to explore. Algorithmically, we think it would be interesting to explore methods for faster sampling, such as adaptive HMC. Empirically, we think it would be interesting to explore, extend, and understand results we've found, in directions such as compositionality, out-of-distribution detection, adversarial robustness, and online learning. Furthermore, we think it may be interesting to apply EBMs on other domains, such as text and as a means for latent representation learning.

# 9 Acknowledgements

We would like to thank Ilya Sutskever, Alec Radford, Prafulla Dhariwal, Dan Hendrycks, Johannes Otterbach, Rewon Child and everyone at OpenAI for helpful discussions.

## Footnotes

*Work done at OpenAI

*Correspondence to: yilundu@mit.edu

*Additional results, source code, and pre-trained models are available at https://sites.google.com/view/igebm

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
