[Supplementary Material]

# Implicit Generation and Modeling with Energy-Based Models

**Yilun Du** *
MIT CSAIL

**Igor Mordatch**
Google Brain

## A    Appendix

### A.1    Additional Qualitative Evaluation

Figure 1: MCMC samples from conditional CIFAR-10 energy function

We present qualitative images from conditional generation on CIFAR10 in Figure 1 and from conditional generation of ImageNet128x128 in Figure 2.

We provide further images of cross class conversions using a conditional EBM model in Figure 3. Our model is able to convert images from different classes into reasonable looking images of the target class while sometimes preserving attributes of the original class.

Figure 2: MCMC samples from conditional ImageNet128x128 models

| | Deer | | Car | | Frog |
|---|---|---|---|---|---|
| | Bird | | Airplane | | Truck |
| | Frog | | Ship | | Ship |
| | Ship | | Truck | | Deer |

Figure 3: Illustration of more cross class conversion applying MCMC on a conditional EBM. We condition on a particular class but is initialized with an image from a another class(left). We are able to preserve certain aspects of the image while altering others

We analyze nearest neighbors of images we generate in L2 distance Figure 4 and in Resnet-50 embedding space in Figure 5.

## A.2 Test Time Sampling Process

We provide illustration of image generation from conditional and unconditional EBM models starting from random noise in Figure 6 with small amounts of random noise added. Dependent on the image generated there is slight drift from some start image to a final generated image. We typically observe that as sampling continues, much of the background is lost and a single central object remains.

We find that if small amounts of random noise are added, all sampling procedures generate a large initial set of diverse, reduced sample quality images before converging into a small set of high probability/quality image modes that are modes of images in CIFAR10. However, we find that if

(a) Nearest neighbor images in CIFAR10 for conditional energy models (leftmost generated, seperate class per row).

(b) Nearest neighbor images in CIFAR10 for unconditional energy model (leftmost generated)

Figure 4: Nearest neighbor images $L2$ distance for images generated from implicit sampling.

sufficient noise is added during sampling, we are able to slowly cycle between different images with larger diversity between images (indicating successful distribution sampling) but with reduced sample quality.

Due to this tradeoff, we use a replay buffer to sample images at test time, with slightly high noise then used during training time. For conditional energy models, to increase sample diversity, during initial image generation, we flip labels of images early on in sampling.

## A.3 Likelihood Evaluation And Ablations

To evaluate the likelihood of EBMs, we use AIS [Neal, 2001] and RAISE to obtain a lower bound of partition function [Burda et al., 2015]. We found that our energy landspaces were smooth and gave sensible likelihood estimates across a range of temperatures and so chose the appropriate temperature that maximized the likelihood of the model. When using these methods to estimate the partition function on CIFAR-10 or ImageNet, we found that it was too slow to get any meaningfull partition function estimates. Specifically, we ran AIS for over 300,000 chains (which took over 2 days of time) and still a very large gap between lower and upper partition function estimates.

While it was difficult to apply on CIFAR-10, we were able to get lower differences between upper and lower partition functions estimates on continuous MNIST. We rescaled MNIST and to be between 0 and 1 and added 1/256 random noise following [Uria et al., 2013]. Table 7 provides a table of log likelihoods on continuous MNIST across Flow, GAN, and VAE models as well as well as a comparison towards using PCD as opposed to a replay buffer to train on continuous MNIST. We find that the replay buffer is essential to good generation and likelihood, with the ablation of training with PCD instead of replay buffer getting significantly worse likelihood. We further find that EBMs appear to compare favorably to other likelihood models.

(a) Nearest neighbor images in CIFAR10 for conditional energy models (leftmost generated, seperate class per row).

(b) Nearest neighbor images in CIFAR10 for unconditional energy model (leftmost generated)

Figure 5: Nearest neighbor images ResNet-50 distance for images generated from implicit sampling.

(a) Illustration of implicit sampling on conditional EBM of CIFAR-10

(b) Illustration of implicit sampling on an unconditional model on CIFAR-10

Figure 6: Generation of images from random noise.

## A.4  Hyper-parameter Sensitivity

Empirically, we found that EBM training under our technique was relatively insensitive to the hyper-parameters. For example, Table 9 shows log likelihoods on continuous MNIST across several different order of magnitudes of L2 regularization and step size magnitude. We find consistent likelihood and good qualitative generation across different variations of L2 coefficient and step size magnitude and observed similar results in CIFAR-10 and Imagenet. Training is

| Models | Parameters | Training Time | Sampling Time |
|---|---|---|---|
| EBM | 5M | 48 | 3 Hour (Variable) |
| PixelCNN++ | 160M | 1300 | 72 Hour |
| Glow | 115M | 1300 | 0.5 Hour |
| SNGAN | 5M | 9 | 0.02 Hour |

Figure 8: Comparison of parameters, training time (GPU hours), and sampling time (for 50000 images) on CIFAR-10. For EBM, sampling time depends on steps of sampling. We used 3 hours of sampling to generate quantitative metrics, but sampling can be much faster (around 0.2 hour) with reduced diversity.

| Model | Lower Bound | Upper Bound |
|---|---|---|
| EBM + PCD | 380.9 | 482 |
| GAN 50 [Wu et al., 2016] | 618.4 | 636.1 |
| VAE 50 [Wu et al., 2016] | 985.0 | 997.1 |
| NICE [Dinh et al., 2014] | **1980.0** | 1980.0 |
| EBM + Replay Buffer | 1925.0 | **2218.3** |

Figure 7: Log likelihood in Nats on Continuous MNIST. EBMs are evaluated by running AIS for 10000 chains

| Hyper-parameter | Value | Lower Bound | Upper Bound |
|---|---|---|---|
| L2 Coefficient | 0.01 | 1519 | 2370 |
| | 0.1 | 1925 | 2218 |
| | 1.0 | 1498 | 2044 |
| Step Size | 10.0 | 1498 | 2044 |
| | 100.0 | 1765 | 2309 |
| | 1000.0 | 1740 | 2009 |

Figure 9: Log likelihood in Nats on Continuous MNIST under different settings of the L2 penalty coefficient and Langevin Step Size evaluated after running AIS and RAISE for 10000 chains. Lower and upper bound in likelihood remain relatively constant across several different order of magnitude of variation

insensitive to replay buffer size (as long as size is greater than around 10000 samples).

## A.5   Comparison With Other Likelihood Models

We compare EBMs to other generative models in Figure 8 on CIFAR-10. EBMs are faster to train than other likelihood models, with fewer parameters, but are more expensive than GAN based models (due to Langevin dynamics sampling), and slower to sample. Training time for PixelCNN++ and Glow are from reported values in their papers, while sampling time and parameters were obtained from released code repositories. We have added the table to the appendix of the paper and added discussion on these trade-offs and intractability of likelihood evaluation in the main paper.

## A.6   Image Saturation

When EBMs are run for a large number of sampling steps, images appear in increased saturation. This phenomenon can be exampled by the fact that many steps of sampling typically converge to high likelihood modes. Somewhat unintuitively, as seen also by out of distribution performance of likelihood models, such high likelihood modes on likelihood models trained on real datasets are often very texture based and heavily saturated. We provide illustration of this phenomenon on Glow in Figure 10.

High Temperature GLOW Samples      Low Temperature GLOW Samples

Figure 10: Low Temperature (High likelihood mode) vs High Temperature (Low Likelihood mode) in Glow Model

## A.7   Details on Continual Learning Training

To train EBM models on the continual learning scenario of Split MNIST, we train an EBM following Algorithm 1 in the main body of the paper. Initially, negative sampling is done with labels of the digits 0 and 1. Afterwards, negative sampling is done with labels of the digits 2 and 3 and so forth.

Figure 11: Illustration of test time generation of various combinations of two independently trained EBMs conditioned on latents on gender, hair color, attractiveness, and age on CelebA.

Simultaneously, we train EBMs on ground truth image label annotations. We maintain a replay buffer of negative samples to enable effective training of the EBM.

### A.8 KL Term

In cases of very highly peaked data, we can further regularize $E$ such that $q$ matches $p$ by minimizing KL divergence between the two distributions:

$$\mathcal{L}_{\text{KL}}(\theta) = \text{KL}q_\theta p = \mathbb{E}_{\tilde{\mathbf{x}} \sim q_\theta} \left[ \bar{E}(\tilde{\mathbf{x}}) \right] + \mathcal{H}\left[ q_\theta \right] \qquad (1)$$

Where $\bar{E}$ is treated as a constant target function that does not depend on $\theta$. Optimizing the above loss requires differentiating through the Langevin dynamics sampling procedure of (??), which is possible since the procedure is differentiable. Intuitively, we train energy function such that a limited number of gradient-based sampling steps takes samples to regions of low energy. We only use the above term when fine-tuning combinations of energy functions in zero shot combination and thus ignore the entropy term.

The computation of the entropy term $\mathcal{H}\left[ q_\theta \right]$ can resolved by approaches [Liu et al., 2017] propose an optimization procedure where this term is minimized by construction, but rely on a kernel function $\kappa(\mathbf{x}, \mathbf{x}')$, which requires domain-specific design. Otherwise, the entropy can also be resolved by adding a IAF [Kingma et al., 2016] to map to underlying Gaussian through which entropy can be evaluated.

### A.9 Additional Compositionality Results

We show additional compositionality results on the CelebA dataset. We trained seperate conditional EBMs on the latents attractiveness, hair color, age, and gender. We show different combinations of two conditional models in Figure 11.

### A.10 Model

We use the residual model in Figure 12a for conditional CIFAR-10 images generation and the residual model in Figure 12b for unconditional CIFAR10 and Imagenet images. We found unconditional models need additional capacity. Our conditional and unconditional architectures are similar to architectures in [Miyato et al., 2018].

We found definite gains with additional residual blocks and wider number of filters per block. Following [Zhang et al., 2019, Kingma and Dhariwal, 2018], we initialize the second convolution of residual block to zero and a scalar multiplier and bias at each layer. We apply spectral normalization on all weights. When using spectral normalization, zero weight initialized convolution filters were instead initialized from random normals with standard deviations of $1^{-10}$ (with spectrum normalized to be below 1). We use conditional bias and gains in each residual layer for a conditional model. We found it important when down-sampling to do average pooling as opposed to strided convolutions. We use leaky ReLUs throughout the architecture.

We use the architecture in Figure 2 for generation of conditional ImageNet32x32 images.

### A.11 Training Details and Hyperparameters

For CIFAR-10 experiments, we use 60 steps of Langevin dynamics to generate negative samples. We use a replay buffer of size of 10000 image. We scale images to be between 0 and 1. We clip gradients to have individual value magnitude of less than 0.01 and use a step size of 10 for each gradient step of Langevin dynamics. The L2 loss coefficient is set to 1. We use random noise with standard deviation

**(a) Conditional CIFAR-10 Model**

| |
|---|
| 3x3 conv2d, 128 |
| ResBlock down 128 |
| ResBlock 128 |
| ResBlock down 256 |
| ResBlock 256 |
| ResBlock down 256 |
| ResBlock 256 |
| Global Sum Pooling |
| dense → 1 |

**(b) Unconditional CIFAR-10 Model**

| |
|---|
| 3x3 conv2d, 128 |
| ResBlock down 128 |
| ResBlock 128 |
| ResBlock 128 |
| ResBlock down 256 |
| ResBlock 256 |
| ResBlock 256 |
| ResBlock down 256 |
| ResBlock 256 |
| ResBlock 256 |
| Global Sum Pooling |
| dense → 1 |

**(c) Conditional ImageNet32x32 Model**

| |
|---|
| 3x3 conv2d, 128 |
| ResBlock down 256 |
| ResBlock 256 |
| ResBlock down 512 |
| ResBlock 512 |
| ResBlock down 1024 |
| ResBlock 1024 |
| Global Sum Pooling |
| dense → 1 |

**(d) Conditional ImageNet128x128 Model**

| |
|---|
| 3x3 conv2d, 64 |
| ResBlock down 64 |
| ResBlock down 128 |
| ResBlock down 256 |
| ResBlock down 512 |
| ResBlock down 1024 |
| ResBlock 1024 |
| Global Sum Pooling |
| dense → 1 |

$\lambda = 0.005$. CIFAR-10 models are trained on 1 GPU for 2 days. We use the Adam Optimizer with $\beta_1 = 0.0$ and $\beta_2 = 0.999$ with a training learning rate of $10^{-4}$. We use a batch size during training of 128 positive and negative samples. For both experiments, we clip all training gradients that are more than 3 standard deviations from the 2nd order Adam parameters. We use spectral normalization on networks. For ImageNet32x32 images, we an analogous setup with models are trained for 5 days using 32 GPUs. For ImageNet 128x128, we use a step size 100 and train for 7 days using 32 GPUs.

For robotic simulation experiments we used 10 steps of Langevin dynamics to generate negative samples, but otherwise use identical settings as for image experiments.

## A.12 Tips And Failures

We provide a list of tips, observations and failures that we observe when trying to train energy based models. We found evidence that suggest the following observations, though in no way are we certain that these observations are correct.

We found the following tips useful for training.

- We found that EBM training is most sensitive to MCMC transition step sizes (though there is around 2 to 3 order of magnitude that MCMC transition steps can vary).

- We found that that using either ReLU, LeakyReLU, or Swish activation in EBMs lead to good performance. The Swish activation in particular adds a noticeable boost to training stability.

- When using residual networks, we found that performance can be improved by using 2D average pooling as opposed to transposed convolutions

- We found that group, layer, batch, pixel or other types of normalization appeared to significantly hurt sampling, likely due to making MCMC steps dependent on surrounding data points.

- During a typical training run, we keep training until the sampler is unable to generate effective samples (when energies of proposal samples are much larger than energies of data points from the training data-set). Therefore, to extend training, the number of sampling steps to generate a negative sample can be increased.

- We find a direct relationship between depth / width and sample quality. More model depth or width can easily increase generation quality.

- When tuning noise when using Langevin dynamics, we found that very low levels of noise led to poor results. High levels of noise allowed large amounts of mode exploration initially but quickly led to early collapse of training due to failure of the sampler (failure to explore

Figure 13: Relative energy of points sampled from $q(x)$ compared to CIFAR-10 train data points. We find that $q(x)$ exhibits a similar distribution to $p_d(x)$ and thus is similar to $p(x)$.

modes). We recommend keeping noise fixed at 0.005 and tune the step size per problem (though we found step sizes of around 10-100 work well).

We also tried the approaches below with the relatively little success.

- We found that training ensembles of energy functions (sampling and evaluating on ensembles) to help a bit, but was not worth the added complexity.
- We found it difficult to apply vanilla HMC to EBM training as optimal step sizes and leapfrog simulation numbers differed greatly during training, though applying adaptive HMC would be an interesting extension.
- We didn't find much success with adding a gradient penalty term as it seems to hurt model capacity.
- We tried training a separate network to help parameterize MCMC sampling but found that this made training unstable. However, we did find that using some part of the original model to parameterize MCMC (such as using the magnitude to energy to control step size) to help performance.

## A.13 Relative Energy Visualization

In Figure 13, we show the energy distribution from $q(x)$ and from $p_d(x)$. We see that both distributions match each other relatively closely, providing evidence that $q(x)$ is close to $p(x)$

## Footnotes

*Work done at OpenAI