[Reviews · NeurIPS 2019]

Reviewer 1



Originality: Many, if not, all of the techniques have been previously proposed. However, to the best of my knowledge, combining these techniques to scale energy-based models to modern deep network architectures is a novel contribution. Energy-based models are far less popular than other forms of generative models. While some recent energy-based approaches exist (e.g. auto-regressive energy machines), this paper demonstrates a new degree of empirical success. Quality: The paper is thorough. The authors present a comprehensive set of experiments, with multiple downstream applications. Various metrics are evaluated for quantitative comparisons in each case. This is above and beyond what is expected. The one aspect that I feel is lacking is a larger discussion about the drawbacks of this approach. In the supplementary material, the authors discuss the training time and inability to evaluate log-likelihoods. It would be helpful if some of this discussion also appeared in the main paper. Clarity: Overall, the paper is clear. The discussion of energy-based models and training techniques (Section 3) is clear and helpful. Some additional technical details would have been useful. For instance, the paper would benefit from a comparison of the number of parameters, sampling time, training time, etc. Some details regarding the experiments were also unclear, particularly the set-up / training procedure for the robot hand trajectory experiments. I’m also not very familiar with the adversarial examples literature, and I found the adversarial robustness section (4.3) somewhat difficult to follow. Significance: The empirical demonstration of good generative performance with energy-based models on modern image datasets is significant. This may help to re-open this family of models as another option for generative modeling. Currently, these models are not very popular within the community. The other most significant aspect of this paper is the demonstration of improved out-of-distribution evaluation as compared with other methods. This is a recently discovered phenomenon, and the fact that these models do not suffer as much as other families of models could open directions for further study. --- Updates: I'm satisfied with the author response and increase my score to 9.

Reviewer 2



Originality: EBMs are a relatively less popular class of generative models when compared to other model families such as VAEs or GANs. Although EBMs themselves are not new, the authors proposed the use of Langevin dynamics for training as well as a sample replay buffer for stabilization, which was an original contribution. The authors’ insights into tips/tricks for stabilizing training was also a nice combination of existing techniques. Quality: The extensiveness of the empirical results demonstrated the high quality of the paper. The authors showed that their approach outperformed or were comparable to state of the art GAN and autoregressive models (e.g. SNGAN) based on FID/IS, achieved SOTA on adversarial robustness, were able to perform trajectory modeling, did well on an online learning task, and showed experiments with compositional generation. Clarity: The paper was relatively clear and easy to follow, barring minor typos/grammatical errors (a few of which I have listed in the Improvements section). Significance: I expect that the contributions of this paper will be very beneficial to the generative modeling community and will open up a new avenue for research in exploring the uses of EBMs in generative modeling, so this paper has high significance. ------------------------------- UPDATE: I appreciate the authors' responses to my questions, particularly with regards to highlighting the role of the online learning experiments in the paper and tying everything together with a better discussion. I will keep my decision to accept.

Reviewer 3



Quantitative evaluation of mode coverage using experiments on augmented MNIST, similar to Metz et al. (2017, Unrolled Generative Adversarial Networks), better shows the quality of the trained energy function (rather than qualitative evaluations). Why in the inpainting and salt and pepper experiments, some of the recovered images are completely different from the given images. For example, in the last row, the recovered plane is a different plane. Moreover, it seems that the recovered images have saturated colors (white background on row three and five, and saturated blue on the last row). Although the code is not provided with the submission, I saw the repository online and played with provided trained model and inference procedure. I think the saturated colors are an important problem of the proposed algorithm. I would like to hear the authors' comments on that. The previous methods by Xie et al. (2016, A Theory of Generative ConvNet) and Ingraham et al. (2019, Learning Protein Structure with a Differentiable Simulator), which use similar Langevin dynamics for sampling from energy-based models should be addressed in the paper. Formatting issue/typo: -Some figures and tables do not have a figure or table number associated with them, while they are referred to in the text with their numbers. -The column's title says "Salt and paper" === I've read and considered the author feedback and other reviews.

[Author Response · NeurIPS 2019]

1. We thank the reviewers for their feedback. We plan to release source code for all experiments done in the paper.

2. **Comparison to Other Models / Drawbacks (R1)** We compare EBMs to other generative models in Figure 1 on CIFAR-10. EBMs are faster to train than other likelihood models, with fewer parameters, but are more expensive than GAN based models (due to Langevin dynamics sampling), and slower to sample. Training time for PixelCNN++ and Glow are from reported values in their papers, while sampling time and parameters were obtained from released code repositories. We have added the table to the appendix of the paper and added discussion on these trade-offs and intractability of likelihood evaluation in the main paper.

**Formatting (R1, R3)** Following R1, R3's comments, we have added captions to all figures.

**Clarifications on Experiments (R1)** Following R1's suggestion, we have added a more detailed description of the experimental setup for each task from section 4-7 in the appendix of the paper. We further plan to release source code for every section to further clarify experiments, as well as pretrained models.

| Models | Parameters | Training Time | Sampling Time |
|---|---|---|---|
| EBM | 5M | 48 | 3 Hour (Variable) |
| PixelCNN++ | 160M | 1300 | 72 Hour |
| Glow | 115M | 1300 | 0.5 Hour |
| SNGAN | 5M | 9 | 0.02 Hour |

Figure 1: Comparison of parameters, training time (GPU hours), and sampling time (for 50000 images) on CIFAR-10. For EBM, sampling time depends on steps of sampling. We used 3 hours of sampling to generate quantitative metrics, but sampling can be much faster (around 0.2 hour) with reduced diversity.

**Minor Issues (R1, R2, R3)** Following R1, we have changed the Helmholtz Machine reference to RBMs/Boltzmann machines in our paper. We have fixed the spelling issues pointed out by both R2 and R3.

**Online Learning/Discussion (R2)** Our motivation to include online learning experiments is the same as our motivation to include out-of-distribution and adversarial robustness experiments: current deep learning models exhibit a number of peculiar failure modes. Susceptibility to adversarial perturbations and distribution shift are two, but catastrophic forgetting is another. Excitingly, we are finding that energy-based models don't seem to be as susceptible to these failure modes. Although a more detailed study on each of these topics is needed, we hope that by highlighting all these intriguing results (including online learning), we can better stimulate the machine learning community to conduct further research on EBMs. But if the reviewers still feel that online learning results are superfluous, please let us know in the meta-review and we will be happy to shorten or remove them.

Following R2's suggestion, we have added an additional paragraph of discussion on future work. Algorithmically, we think it would be interesting to explore methods for faster sampling, such as adaptive HMC, as well as other techniques such as score/moment matching training of EBMs. Empirically, we think it would be interesting to explore, extend, and better understand results we've found, in directions such as compositionality, out-of-distribution detection, adversarial robustness, and online learning. Furthermore, we think it may be interesting to apply EBMs on other domains, such as text, where predominant models are likelihood based and as a means for representation learning.

**Historical Ensembles (R2)** We find that generation diversity appears to improve logarithmicly with the number of ensembles, with marginal gains above an ensemble of 10 models. We have add this to the paper.

**Mode Evaluation (R3)** Following R3, we evaluated the number of modes under the stacked MNIST task. We obtained all 1000 modes with a KL divergence of 0.79, which compares favorably to [Metz et al., 2016]. However, such quantitative metrics do not measure mode coverage well in EBMs, as they measure the probability distribution of the sampler as opposed to that of an EBM. Sampling with finite steps of MCMC can be biased towards certain images and completely ignore other high likelihood modes of the EBM. Instead, we believe our qualitative evaluation of mode coverage allows mode coverage even in regions that are difficult for a sampler to generate. Furthermore our compositionality results, where we are able to combine independent conditional EBMs to generate precise dSprites images, show that each conditional model has modes at all 737280 combinations in dSprites.

**Saturation/Image Recovery (R3)** We illustrated the worst case recoveries of images in the bottom row of the figure 4b in the paper. In such scenarios, gradients for MCMC sampling may be biased towards to a different image, leading to an overall image change. However, this does not imply a lack of a mode at the correspond image, as when the model is initialized with a ground truth image, the image is maintained. The saturation effect in the corresponding images is due to sampling converging to high likelihood modes. Such high likelihood modes are saturated images because these saturated images are smoother and thus more explainable. This phenomenon is a ubiquitous trait of many likelihood based deep generative models. We illustrate in Figure 2 the same effect in GLOW, where lower temperature samples (high likelihood images) (right) and more saturated than higher temperature samples (low likelihood images) (left). We have added these clarifications to the paper.

High Temperature GLOW Samples     Low Temperature GLOW Samples

Figure 2: Low Temperature (High likelihood mode) vs High Temperature (Low Likelihood mode) in Glow Model

**Related Work (R3)** We have addressed and added the related work suggested by R3 in the paper. Xie et al. [2016] applies Langevin sampling on simple datasets of up to 4 images. Ingraham et al. applies Langevin sampling on the separate domain of protein folding, and add multiple explicit losses on sampling as well as an additional refinement network on top of sampling to allow effective generation.

[Meta-Review · NeurIPS 2019]

This work presents one of the more significant empirical advances in energy-based modeling for image data. The results on up to 128x ImageNet are good and I expect many future works will continue to build on these ideas to sufficiently train energy-based models and leverage their architectures.